# Musical Training for Auditory Rehabilitation in Hearing Loss

**DOI:** 10.3390/jcm9041058

**Published:** 2020-04-08

**Authors:** Jacques Pesnot Lerousseau, Céline Hidalgo, Daniele Schön

**Affiliations:** Institut de Neurosciences des Systèmes, Aix Marseille University, Inserm, INS, 13005 Marseille, France; celine.hidalgo@univ-amu.fr (C.H.); daniele.schon@univ-amu.fr (D.S.)

**Keywords:** musical training, hearing loss, cochlear implants, speech development

## Abstract

Despite the overall success of cochlear implantation, language outcomes remain suboptimal and subject to large inter-individual variability. Early auditory rehabilitation techniques have mostly focused on low-level sensory abilities. However, a new body of literature suggests that cognitive operations are critical for auditory perception remediation. We argue in this paper that musical training is a particularly appealing candidate for such therapies, as it involves highly relevant cognitive abilities, such as temporal predictions, hierarchical processing, and auditory-motor interactions. We review recent studies demonstrating that music can enhance both language perception and production at multiple levels, from syllable processing to turn-taking in natural conversation.

## 1. Introduction

Before the advent of text messages and email, speech was undoubtedly the most common means of communication. Yet, even nowadays, speech or verbal communication seems to be the most efficient and easier way of conveying thoughts. Speech uses many combinations of phonetic units, vowels, and consonants to convey information. These units can be distinguished because each has a specific spectro-temporal signature. Nonetheless, one may wonder which acoustic features are fundamental for speech perception. Surprising as it may seem, the answer is that there probably are no necessary acoustic features for speech perception. This is because speech perception is an interplay between top-down linguistic predictions and bottom-up sensory signals. Sine-wave speech, for instance, although discarding most acoustic features of natural speech, except the dynamics of vocal resonances, can still be intelligible [1]. If speech perception indeed depended upon the specific spectro-temporal features of consonants and vowels, then a listener hearing sinusoidal signals should not perceive words. This is similar to the fact that *it deosn’t mttaer in waht oredr the ltteers in a wrod are, the olny iprmoetnt tihng is taht the frist and lsat ltteer be at the rghit pclae. The rset can be a toatl mses and you can sitll raed it wouthit porbelm. Tihs is bcuseae the huamn mnid atnicipaets the inoframtion folw.* A predictive coding framework minimizing the error between the sensory input and the predicted signal provides an elegant account of how prior knowledge can radically change what we hear.

This perspective is relevant in the study of the potential benefits of music-making to speech and language abilities. First, because both music and language share the distinctive feature of being dynamically organized in time, they require temporally resolved predictions. Second, in the predictive coding perspective, music and language share the same universal computations, despite surface differences. The similarity might, therefore, exist at the algorithmic level even though specific implementation may differ. Although a view of language and music has dominated the last century as two highly distinct human domains, each with a different neural implementation (modularity of functions), the current view is rather different. To perceive both music and language, one needs to be able to discriminate sounds. Sounds can be characterized in terms of a limited number of spectral features, and these features are relevant to both musical and linguistic sounds. Both linguistic and musical sounds are categorized. This is important to make sense of the world by reducing its intrinsic variety to a finite and limited number of categories. However, we typically perceive sounds in a complex flow. This requires building a structure that evolves in time, considering the different elements of the temporal sequence. Our previous experience with these sounds will heavily influence the way sounds are perceived in a structure. Such experience generates internal models that allow us to make accurate predictions on upcoming events. These predictions can be made at different temporal scales and affect phoneme categorization, semantic, syntactic, and prosodic processing.

Once ascertained that music and language share several cognitive operations, one can wonder whether music training affects the way the brain processes language, and vice-versa. Indeed, if some of the operations required by music are also required by language, then one should be able to observe more efficient processing in musicians compared to nonmusicians whenever the appropriate language processing levels are investigated. Overall, there has been an increasing number of studies in the last decades, pointing to an improvement induced by music training at different levels of speech and language processing [2]. However, a debate on these effects remains open, especially at the high processing levels. The reason for this success seems to rely on, besides the sharing of sensory or cognitive processing mechanisms, the fact that music places higher demands on these mechanisms compared to speech [3]. This is particularly evident when considering pitch processing, temporal processing, and auditory scene analysis. Music-making requires a high level of both temporal and spectral prediction by adopting a prediction perspective. Indeed, music-making requires a highly precise level of temporal synchronization. It also requires segregating the sound of similar instruments, e.g.; the viola and the violin in a string quartet, which is only possible with an accurate prediction of the spectral content of the music. Importantly, these findings showing a benefit of musical training at different levels of speech and language processing have been underpinned by differences in terms of neural structures and dynamics between musicians and nonmusicians [4]. Altogether, the similarities between music and language in terms of cognitive operations and neural implementation, coupled to music-induced superior language skills, have provided a solid ground for the use of music in speech and language disorders. For reviews see [2,5,6].

When turning to hearing-impaired people, one should acknowledge that, despite early screenings, technology improvements, and intense speech therapy remediation, they still generally suffer from specific deficits in language comprehension and production. Without being exhaustive, children with cochlear implants (CI) and adults with hearing impairments have deficits in discriminating spectral features [7,8], phoneme categorization [9], and perception of speech in noise [8,10,11,12,13]. Beyond problems directly related to the implant’s limitations in terms of spectral resolution, children with CI also suffer from a lack of auditory stimulation during the first months of life, which possibly leads to deficient structural connectivity [14] and deleterious cortical reorganization [15,16,17].

The aim of speech therapy (with children) is to reorient these pathological developmental trajectories. The beginning of a speech therapy usually consists of focusing on low-level features of speech, such as phoneme recognition or spectro-temporal pattern recognition. Later, more integrated features are progressively added, such as syntactic complexity, mental completion, and implicit statements comprehension. However, although benefiting from stimulation of language-related cognitive processes at different levels, hearing-impaired people rarely reach normal-level performance. For reviews, see [18] on the effects of auditory verbal therapy for hearing-impaired children; in [19] on computer-based auditory training for adults with hearing loss. Considering the vast literature on the benefits provided by intense musical training in normal hearing people, a new line of research has emerged to propose new therapies to hearing-impaired people. Although no studies have carefully contrasted standard speech therapy with musical training, the evidence is accumulating in favor of the latter. By providing complex auditory stimuli and tight interactions between perception and action, musical training is thought to enhance top-down auditory processing and induce brain plasticity at multiple levels, even at shallow ones [20].

Indeed, music training has become increasingly common in hearing impairment rehabilitation, especially in children. Following this growing interest, new scientific literature has flourished and begun to examine the potential benefits that music training provides. We present here a review of most works assessing the potential benefit of musical training on hearing-impaired people (Figure 1). These results are organized concisely according to the level in the hierarchy of the auditory processes they address from spectro-temporal discrimination to the perception of speech in noise (Figure 2). For extensive reviews, we draw the attention of the reader toward two recent reviews on the topic, one for children with CI [21] and one for elderly adults [22]. Although we share the enthusiasm for this line of research. We would like to raise a note of caution on the fact that: (1) most studies are correlational, thus obliviating definitive causal statements [23]. (2) Reported effects are generally moderate and (3) rigorous scientific methodology is sometimes hard to achieve in clinical settings with small samples and important inter-individual variability. Randomized controlled trials, with big samples of hearing-impaired participants randomly assigned to experimental and control groups, are still lacking and would be highly valuable to estimate musical training effects accurately. Despite these issues, encouraging results are provided in the literature, and musical training offers significant potential for future auditory remediation therapies.

## 2. Discriminate Sounds Based on Pitch, Duration, and Timbre

At a very rapid temporal scale, speech can be described by the fundamental characteristics of its constituting sounds: pitch, duration, and timbre. These features are critically related to the phonemic level of language, and as such, constitute a privileged target for hearing impairment remediation.

### 2.1. Pitch

Pitch, measured in the number of oscillatory cycles per second (Hz), is an important cue for both music and language. In speech, pitch relates to the notion of the fundamental frequency (F0), i.e.; the frequency of the vocal cords’ vibration. F0, as well as sound harmonics F1, F2, etc.; are important cues to identify vowels, tones, sex of the locutor, and information conveyed by prosody. In tone languages, pitch variations also play a role in distinguishing words and grammatical categories. In music, pitch differentiates a high from a low note, and as such is the first constituent of melody. Importantly, small pitch variations, e.g.; 6% for a halftone, are highly relevant in music, while this is not the case in speech.

It has been shown that musicians have better pitch discrimination abilities than non-musicians in controlled psychophysical studies using pure tones [49,50,51,52]. This advantage is correlated to a better ability to discriminate speech tones in adult musicians [53,54,55], as well as in 8-year-old children practicing music [56]. This advantage extends to vowel discrimination in natural speech, even after controlling for attention [57]. Neural correlates of this difference between musicians and non-musicians have been found in the brainstem response to sounds. Measured with an electroencephalograph (EEG), brainstem responses are acquired via multiple repetitions of a simple short stimulus, such as a syllable. As the brainstem response mimics the acoustic stimulus in both intensity, time, and frequency [58], a measure of acoustic fidelity can be established. It has been shown that musicians have a higher acoustic fidelity in their brainstem response than nonmusicians [59]. This advantage can be naturally interpreted as a better encoding of the stimulus, leading to a better perception. However, it is also important to keep in mind that it may also reflect better prediction abilities, and stronger top-down connections between the cortex and the brainstem, notably via the cortico-fugal pathway [60].

Concerning concrete enhancement of pitch discrimination abilities in hearing-impaired people, musical training is effective for elderly adults, as well as for children with CI. For elderly adults, one correlational study [37] showed a musician’s resilience to age-related delays in the neural timing of syllable subcortical representation. For children with CI, two studies reported differences in F0 discrimination between children having or not having a musical family environment [42]. Children living in a singing environment showed larger amplitude, and shorter latency of a P3a evoked potential, following changes in F0, indicating a change in the brain dynamics. Another study reported an enhancement of F0 discrimination following 2–36 months of musical schooling [61]. Finally, one valuable intervention study reported the same enhancement after two school terms of weekly musical training, although not related to speech-in-noise perception amelioration [45].

### 2.2. Duration

The ability to estimate and discriminate sound duration is crucial for recognizing consonants in speech [62]. It constitutes the main contrast between pairs of voiced consonants. For example, the only difference between a “p” and a “b” is the relative timing between the burst onset and the voice onset, a relation called voice onset time (VOT). In music, duration discrimination is relevant to distinguish different rhythmic patterns, as well as a different interpretation of the same melody (staccato vs. legato).

Multiple studies have shown that musicians have better timing abilities. They achieve lower gap detection thresholds [63] and better temporal intervals discrimination [64,65]. These perceptual benefits are linked to speech-related abilities, insofar as musicians are also better at discriminating syllable duration [66]. Similarly to the pitch dimension, these behavioral measures are linked to neural data, especially to the brainstem response consistency. Indeed, the musician’s brainstems encode better rapid transients in speech sounds [67]).

Children with CI suffer from a deficit in duration discrimination abilities, resulting in troubles with phoneme perception [9] and phonological awareness [68,69]. Musical training could, therefore, be a valuable resource for rehabilitation. Indeed, a recent correlational study has investigated the phonemic discrimination abilities of children with CI following 1.5 to 4 years of musical experience [40]). Regression analysis indicated that these children had better scores after training and that musical lessons were partly driving the improvement.

### 2.3. Timbre

Timbre is a more subtle property of sound. It is related to high order properties, such as harmonics and temporal correlations. Two sounds can have the same pitch and duration, but different timbres. For example, the only difference between a note played on a clarinet and the same note played on a piano is their timbre. In both music and speech, timbre is essential for the notion of categorization. Despite differences in timbre, the musician must recognize notes or chords, even though they are played on different instruments. Similarly, a listener must recognize phonemes even if different talkers pronounce them with different voices.

Indeed, musicians have better timbre perception of both musical instruments and voice [70], and reduced susceptibility to the timbral influence of pitch perception [71]. Their brainstem responses to various instrument timbres are also more accurate [72]. Interestingly, this advantage generalizes to language stimulus, such as vowels [73].

Musical training has proven to be effective in children with CI to improve timbre perception. Although not correlated to changes in auditory speech perception, one study has shown improvement in the capacity of recognizing songs, tunes, and timbre in children with CI after an 18 months musical training program [35]. Another study has revealed that the P3a evoked potential to change in timbre was earlier in children living in a singing environment, suggesting that their timbre perception was enhanced [42].

## 3. Exploit the Temporal Structure and Group Sounds Together

The ability to extract fundamental properties of individual sounds is essential for speech comprehension, especially at the phonemic level, and musical training has proven to be effective at enhancing this ability. As such, it is a promising tool for remediation in hearing-impaired people, both for children with CI and old adults. However, speech and music are not reducible to a sum of their constituting sounds. At a slower temporal scale, speech and music are both defined by a hierarchy of embedded rhythms and by multiple levels of chunks.

### 3.1. Temporal Structure

Speech and music are both characterized by a hierarchy of rhythms. Short units, such as phonemes and notes, are embedded into longer units, such as words or melodic phrases. The distribution of the frequency of items at each level is not random. Still, it follows a systematic pattern, reproducible across languages: in speech, phonemes happen around 15–40 Hz (15 to 40 per second), syllables around 4–8 Hz, and words around 1–2 Hz [74]. As non-random processes, these rhythms allow the brain to compute expectations and predictions about when the next sound will come. It has been suggested that the synchronization between brain oscillations and speech rhythm, referred to as “entrainment,” is the primary mechanism allowing temporal predictions in speech, and in fine speech comprehension [75,76]. Indeed, the speech multiple frequencies coincide with natural rhythms of brain activity, especially gamma, theta, and delta rhythms [77]. A similar hierarchical structure is involved in music, i.e.; notes are inside rhythms and rhythms inside longer melodic phrases. Similar brain oscillatory mechanisms subtend the extraction of this temporal structure [78]. Furthermore, extracting the temporal structure is also involved in interpersonal synchronization and group playing, where fine-grained temporal synchronization, accurate temporal predictions, and low temporal jitters between players is needed. From the neural standpoint, music and speech both activate the dorsal pathway of the auditory network [79], especially the interaction between the motor and auditory systems [80].

Similar to low-level aspects of sounds, musicians are better at extracting the temporal structure of an auditory stream. They are generally better in rhythm perception [64], rhythm production [27,81], and audiomotor synchronization to both simple [82,83,84] and complex rhythms [85]. The enhanced ability to track the beat positively correlates with auditory neural synchrony and more precisely with subcortical response consistency [86]. Musicians are also better at performing hierarchical operations, such as grouping sounds in a metrical structure [87,88].

Concerning rehabilitation, there has been a lot of evidence in various clinical populations in favor of rhythmic musical training (for reviews see [5,89]). While music intervention for language disorders has initially been proposed for aphasia [90,91], in particular a music rhythm intervention [92]), in recent years musical interventions have become quite common for children with dyslexia. More precisely, since dyslexia has been characterized as involving a deficit in timing abilities [93], the rhythmic properties of music have been regarded as potentially beneficial to normalize the developmental trajectory [94]. One randomized controlled trial has shown the specific positive effect of 2 years of musical training on phonological awareness in children with a reading disorder [95]. One study showed that 3 years of musical schooling was associated with improvements in phonological awareness and P1/N1 response complex in neurotypical adolescents ([96]; however, see [97]).

For hearing-impaired people, it has been shown that short rhythmic training can enhance the ability to exploit the temporal structure. One study using a priming paradigm showed that rhythmic primes congruent with the metrical structure of target sentences could help children with CI to repeat better phonemes, words, and sentences [26]. A series of two studies have also demonstrated that 30 min of rhythmic training induces a better detection of word rhythmic regularity during verbal exchanges [32] and more precise turn-taking [33] in 5–10 years old children with CI.

### 3.2. Working Memory

Working memory, the ability to temporally maintain and manipulate information [98], is critical for language comprehension [99]. From a general point of view, working memory is important to interpret current items in the light of previous ones, to form chunks and to build expectations about future coming items. Indeed, keeping track of the beginning of a sequence to predict the rest is one fundamental computation for speech comprehension. It has been proven on large meta-analyses that working memory, as both storage and process, is a good predictor (Pearson’s ρ = 0.30 to 0.52) of global speech comprehension [100]. In music, working memory is also strongly involved in both perception and production, e.g.; when keeping track of a chord progression or when memorizing the score online to play on time.

While the ability to anticipate events and learn statistical dependencies between items is enhanced in musicians [101], children with CI seem to have a specific deficit ([102]; however see [103]). Furthermore, both verbal [104] and non-verbal [105] working memory is impaired in children with CI.

Once again, musical training has shown some benefits for working memory in hearing-impaired individuals. Children with CI enrolled in a musical training program for at least one year have a better auditory working memory [7,40]. One intervention study has also shown an enhancement of memory for melodies after six months of piano lessons compared to six months of visual training [29]. In older people, evidence suggests that musicians with at least 10 years of musical training have better verbal [30,36] and non-verbal ([31]; see [106] for a meta-analysis) working memory. However, the reported effects are weak and inconsistent.

## 4. Perceive Speech in Noise, Prosody, and Syntax

### 4.1. Speech in Noise

Speech perception in noise, e.g.; speaking on the phone or in social gatherings, is recognized as having a strong impact on the quality of life and is systematically impaired in hearing-impaired people [8,10,11,12,13]. Although several cognitive processes are involved in perceiving speech in noise, such as attention, memory, and listening skills, the ability to analyze auditory scenes stands first. Auditory scene analysis is defined as the capacity to group sound elements coming from one source (one speaker) and segregate elements arising from other sources (other speakers) [22]. In music, auditory scene analysis is both prevalent and challenging in ensemble playing, for instance, when listening to a string quartet wherein the four voices are intermingled strongly.

There is evidence that musical training provides an advantage in the ability to group and segregate auditory sources [47]. This includes pure tones, harmonic complexes, or even spectrally limited “unresolved” harmonic complexes [107]. Musicians are better at perceiving and encoding speech in difficult listening situations. They better discriminate and encode the F0 and F1 of a vowel in the presence of severe reverberation [108]. They encode speech in noise more precisely in the brainstem response [109,110,111]. This encoding advantage extends to the perception of speech in noise [112]. It has been evaluated to ~0.7 dB of signal-to-noise ratio, which translates to ~5–10% improvement in speech recognition performance [22,113]. However, it should be noted that this point is debated heavily and other results fail to replicate [114,115,116,117,118]. One intervention study has shown that the perception of speech in noise was better in normal-hearing children after two years of musical training compared to 1 year [119]. However, results should be mitigated by the fact that the advantage appeared only after two years, and not after one year.

In hearing-impaired people, although no intervention study has been conducted, several quasi-experimental or cross-sectional studies have provided encouraging results. Children with CI who choose to sing at home have enhanced speech in noise [43] and children with hearing impairment who receive musical training have better auditory scene analysis [40]. These results are to be interpreted with caution, as sample sizes are small (*n* ≈ 15 per group), confounding factors are not thoroughly considered (e.g.; no randomization), and thus, causal interpretation of these results should be avoided. In older people, studies on life-long musicians have shown speech in noise perception [36,47] and auditory scene analysis enhancement [48].

### 4.2. Prosody

Prosody refers to the variation of intensity, duration, and spectral features across time in speech. It is an integrated property, as it concerns the dynamics of the entire phrase, for example, the rise of F0 during a question sentence. It conveys emotions, contextual information, and meaning. It is very close to the melodic aspect of music. Indeed, musicians have better discrimination and identification of emotional prosody [120], and a higher and earlier evoked response to changes in prosody [121]. In children with CI, one study showed that six months of piano lessons was associated with enhanced emotional prosody perception, although not significantly different from a control group who did not receive musical training [29].

### 4.3. Syntax

Finally, at the highest level of analysis, language is composed of embedded syntactic structures [122,123]. Similar to language, music can also be described as having a syntactic structure [124], with its own grammar. For example, in a given context, certain words are grammatically incorrect (* I have *hungry*). Similarly, in a musical sequence, certain chords or notes are highly unexpected (* Bm, G, C, *A#*). For language, these grammatical errors are associated with specific evoked potentials, such as the early left anterior negativity [125] or the P600 [126]. Similar components are evoked by musical “grammatical” errors (ERAN [127]; P600 [128]; for reviews see [129,130]).

Musical training has been effective in restoring evoked potentials associated classically with syntactic processing in various clinical populations. For example, the P600 component is present after three minutes of rhythmic priming in patients with basal ganglia lesions [131]). In contrast, it was proven absent in this type of patient [132]. Similarly, rhythmic primes have been beneficial for grammaticality judgment in normal children, as well as in children with specific language impairment [133,134]. In children with hearing impairment, only one study has been conducted. It suggests a positive, although moderate, effect of rhythmic primes on grammatical judgement, possibly mediated by an effect on speech perception [25]. No effect was revealed for syntactic comprehension. In these studies, children were tested immediately after the rhythmic primes. Further work is needed to establish the presence and the duration of any carryover effect. Indeed, to be useful for therapy, these positive effects must last over time, at least in the order of the day.

## 5. Toward a Remediation of Dialogue by Musical Training

Communication and verbal coordination often go along with a largely automatic process of “interactive alignment” [135]. Two individuals talking with each other simultaneously align their neural dynamics at different linguistic levels by imitating each other’s choices of voice quality, speech rate, prosodic contour, grammatical forms, and meanings [136]. Multi-level alignment improves communication by optimizing turn-taking and coordination behaviors. Importantly, communication is a central feature of social behavior. Furthermore, communicating requires being able to predict others’ actions and integrate these predictions in a joint-action model. Joint-action and prediction can be found in several types of human (and animal) behavior. For all these examples, to succeed, it is not enough to recognize the partners’ actions; rather, it is essential to be able to predict them in advance. Failure in doing so will most often imply the inability to coordinate the timing of each other’s actions [137].

Clearly, music-making is highly demanding in terms of temporal precision and flexibility in interpersonal coordination at multiple timescales and across different sensory modalities. Such coordination is supported by cognitive-motor skills that enable individuals to represent joint action goals and to anticipate, attend, and adapt to other’s actions in real-time. Several studies show that music training can have a facilitatory effect at several levels of speech and language processing [3,6,60]. Meanwhile, synchronized activities lead to increased cooperative and altruistic behaviors in adults [138], children [139], and infants [140]. While little is known of the effect of music training on interpersonal verbal coordination and social skills in individuals with HL, the few existing studies [32,33] show that this may be a promising avenue of research.

Overall, hearing devices are developed for rehabilitating language comprehension and production. *In fine*, the long term goal is to improve communication in hearing-impaired people. Music, especially ensemble playing, offers a particularly stimulating social situation. As such, it is a promising tool for developing social aspects of language, such as turn-taking, language level flexibility, role-playing or even joking. These are aspects that have a tremendous impact on the quality of life of hearing-impaired people.

## Figures and Tables

**Figure 1 jcm-09-01058-f001:**
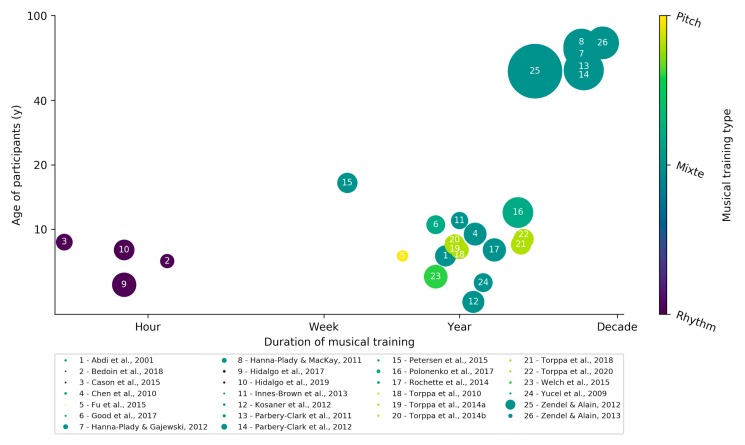
Review of musical training protocols reported in scientific papers [7,24,25,26,27,28,29,30,31,32,33,34,35,36,37,38,39,40,41,42,43,44,45,46,47,48]. Each circle represents one study, plotted as a function of the average duration of the musical training and the average age of the participants. Color of the points indicates the content of the training, on a continuum from rhythmic only (drums only) to pitch only (songs, melodies) training. The size of the points indicates the sample size of the study (range: 6–163).

**Figure 2 jcm-09-01058-f002:**
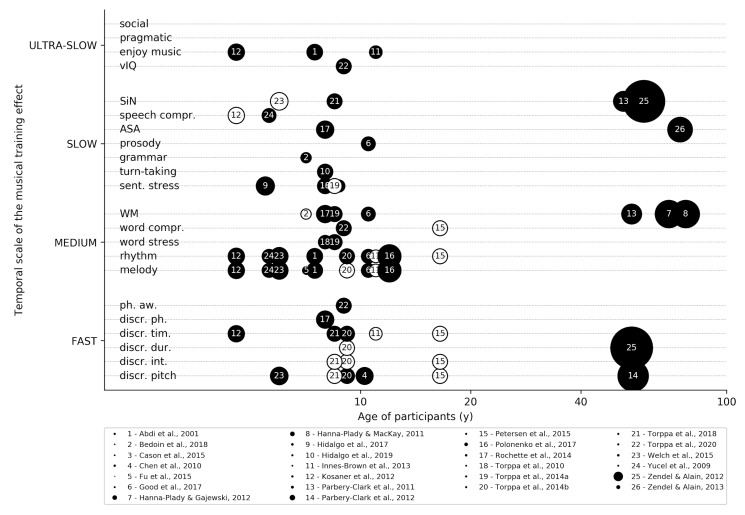
Review of musical training effects for hearing impaired people [7,24,25,26,27,28,29,30,31,32,33,34,35,36,37,38,39,40,41,42,43,44,45,46,47,48]. Each circle represents one study, plotted as a function of the average age of the participants and the precisely measured effect. Black circles: statistically significant effects; white: non-significant. The size of the points indicates the sample size of the study (range: 6–163).

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
