# Peer review of "Musical Training for Auditory Rehabilitation in Hearing Loss"

_jcm, 2020, doi:10.3390/jcm9041058_

Round 1
Reviewer 1 Report
This interesting review article tackles an important, highly relevant topic that might have clinical implications: the extent to which musical training might result in performance enhancement of non-musical skills such as speech perception, speech production, and others.
The authors show comprehensive knowledge of the literature, and the visualization in Figures 1 and 2 represent excellent ways of summarizing a large amount of data.
Having said that, I believe the manuscript missed the opportunity to take a critical look at the relevant literature. In this day and age when the replication crisis is becoming more recognized across scientific fields, it would be good to examine the literature more critically to start figuring out which results are strongly supported, which are more tenuous, and which are simply false positive results possibly obtained through p-hacking practices (in other words, inappropriate use of “investigator degrees of freedom” to analyze the data in many different ways until a “significant” result is found). After all, one of the most famous non-replicated results in the recent past (the “Mozart effect”) falls squarely within the subfield examined in the present manuscript.
I have not examined carefully all the papers cited in this review article, but many of the claims that are made do not line up. In some cases, the original studies were misinterpreted. In others, the original studies committed errors of analysis that were uncritically accepted in the present manuscript. Last but not least, several studies that question some of the review’s conclusions were not cited. In the specific comments that follow I have indicated the most serious or clearest mistakes with three asterisks (***).
It is almost certain that musical training has numerous downstream consequences, and a careful, unbiased analysis of the literature would be an important contribution. Unfortunately, the underlying literature itself seems to be so riddled with misinterpretations, p-hacking, and plain mistakes, that an uncritical compilation of published results is unlikely to shed new light on this important topic.
I realize that many of the following comments may sound harsh and perhaps some of them are incorrect. Nonetheless, I hope the authors are persuaded to take a closer, more critical look at the literature of their field.
SPECIFIC COMMENTS
Lines 40-41, “a predictive coding perspective minimizes 40 the importance that these differences may play at the algorithmic processing level.”
I’m not sure what this means.
Lines 50-52: Sounds reasonable to me.
Lines 58-59: “Overall, in the last decades there has been a massive accumulation of evidence pointing to an improvement induced by music training at different levels of speech and language processing”.
But as we will see below, at least some of this evidence is highly questionable.
Lines 70-73: “Altogether, the similarities between music and language in terms of cognitive operations and neural implementation, coupled to music-induced superior language skills have provided a solid ground for the use of music in speech and language disorders (Fujii & Wan, 2014)”.
Fujii & Wan does not provide direct support to the claim about “music-induced superior language skills”, it is another review article where I could not find original data (maybe I missed it?).
Line 83: “deleterious cortical reorganization (Lee et al., 2001)”.
This classic study is an example of potential p-hacking. The main result is based on only 10 cochlear implant users. The authors say they “examined and compared glucose metabolism in the auditory and related cortices in 15 prelingually deaf patients before cochlear implantation”, but they only present post-implantation data from 10 patients without explanation. They mention two areas of interest (superior temporal and inferior frontal), and they also have three hemispheric choices: left, right, or both hemispheres. Taken together, this gives them six opportunities to obtain a significant result. They also had the additional opportunity to focus only on primary, only secondary, or all auditory cortex. In the end they pick bilateral superior temporal activity as the variable of interest. Because there is no evidence that this choice was made a priori, the actual probability of a false positive result is much greater than the stated p-value. As we will see, this is a problem with many other papers.
Line 87: “suppleance”, I think this is not an English word.
Lines 92-94, “Musical training, by providing complex auditory stimuli and tight interactions between perception and action, is thought to enhance top-down auditory processings and induce brain plasticity at multiple levels, even at very low ones (Chandrasekaran & Kraus, 2010).”
Actually, the cited paper supports the contention that musical training has an effect on FFR responses but no data to support the claim about top-down auditory processing. This is just speculation on the original authors' part, appropriate as it might be.
***Lines 95-100: It is good that the authors mention the Torppa and the Alain reviews, but it is not entirely clear the extent to which the present review goes beyond those two references. I am not saying it doesn’t, just that the case is not clear.
Lines 101-104: the notes of caution are appropriate but, as discussed below, they may not go far enough.
***Lines 233-234: “One study has demonstrated the positive effect of rhythmic priming on dyslexic children (Przybylski et al., 2013)”.
This seems like a complete mischaracterization of the study. There was no group-priming interaction. This means there was no evidence that rhythmic priming helped dyslexic children any more that it helped normal hearing controls. But second, and most important, the only observed effect was that regular musical prime sequences had a positive effect on grammaticality judgments with respect to irregular musical prime sequences. We do not know what performance would have been obtained in the absence of musical priming. It is possible, for example, that regular priming was neutral (or even slightly deleterious) whereas irregular musical priming was even worse. In any case, this study did not show that rhythmic priming has any positive effect with respect to a control condition that does not use priming, because such a control condition was not included in the study.
Yes, I agree that the rhythmic priming effect shown in other studies is very interesting but this particular study is mischaracterized.
***Lines 241-242: “One study using a priming paradigm has shown that rhythmic primes can help children with CI to repeat better phonemes, words and sentences (Cason et al., 2015).”
One important fact is not mentioned here. The very specific type of rhythmic priming that proved to be useful in the study was the “matching” type. As Cason et al. indicate “There was no significant difference between baseline and mismatching conditions at any level”.
An alternative interpretation of Cason et al.’s results is that if children are informed about the stress structure of the sentence to be presented, they are able to reproduce it better.
In any case, the statement that “rhythmic primes can help children with CI to repeat better phonemes, words and sentences” seems misleading because the nonmatching primes were as “rhythmic” as the matching primes and it is not clear that they had any effect on vowel, consonant, word, or sentence recognition at all.
Line 251, “working memory, as both storage and process, is an excellent predictor of global speech comprehension (Daneman & Merikle, 1996)”.
Working memory is indeed a predictor, yes, but to call it an "excellent" one is an overstatement. Correlations ranged from 0.30 to 0.52, even with storage plus process measures (and 0.14 to 0.40 for storage-only measures).
Lines 256-257: It’s nice that two studies are cited and one fails to replicate the other one. This allows the reader to put the study with the statistically significant result into perspective.
Line 257: The claim that verbal working memory is impaired in children with CI (based on Nittrouer et al.’s study) is at least questionable, because in that study the evaluation of working memory was somewhat confounded with speech intelligibility. In other words, if a child mishears a word his measured “memory” score will be affected due to a perceptual rather than a memory limitation.
***Lines 263-265, “musicians that had at least 10 years of musical training have better verbal (Hanna-Pladdy & Gajewski, 2012; Parbery-Clark et al., 2011) and non-verbal (Hanna-Pladdy & MacKay, 2011) working memory”.
This is a fair description of Parbery-Clark et al.’s paper, which also shows an advantage for speech perception in noise.
On the other hand, the Hanna-Pladdy studies paint a mixed picture and those results should be taken with a large grain of salt. To begin with, all significant effect sizes are small (for example, in the 2011 paper effect sizes ranged from 0.08 to 0.13). In the 2012 paper 27 measures were examined and only 5 were significant. In the 2011 study 13 measures were examined and only 4 were significant. And they are not even the same measures. The initial WMS-VRII result (p=0.02) in the 2011 paper did not replicate in the 2012 paper; out of the five significant results in the 2012 paper, three had been tested in 2011 and results were not significant. Moreover, the distribution of p-values across the two studies is left skewed rather than right skewed, suggesting the possible absence of real effects (see “P-Curve: A Key to the File-Drawer”, by Simonsohn et al.). Last but not least, the 2011 paper evaluated nonverbal memory using two tests (WMS-III VRI and VRII): one gave statistically significant results (although only a 0.106 effect size) and the other did not. Summarizing these results as “musicians have better non-verbal working memory” seems excessively optimistic. Even worse: the same two tests were used in the 2012 paper (which had a higher N per group than the 2011 paper) and there were no statistically significant results at all. Taken together, these results raise the possibility that the 2011 VRII result was a false positive. Again, summarizing this situation by stating that musicians have better nonverbal working memory seems excessive, it feels like cherry-picking results that do show an effect.
***Lines 284-285, “This advantage extends to speech perception in the presence of reverberation (Bidelman & Krishnan, 2010)”.
This is just wrong, or at the very least misleading. Bidelman and Krishnan did not even measure speech perception in reverberant environments. The only behavioral measures shown in that paper are difference limens for F0 and F1. These do happen to show a very large difference favoring the musicians’ group, at any level of reverberation. As Bidelman and Krishnan point out, “For both types of discrimination, musicians obtained DLs which were 2–4 times better than non-musicians”. This is a huge effect, and supports the hypothesis that musicians have auditory perceptual advantages in some domains. Unfortunately, there were only four subjects in each group.
Bidelman and Krishnan do make reference to the “Neural basis for musician advantage to hearing in reverberation”, but there is a big problem: their study does not show any behavioral musician advantage that is specific to reverberation. There was no significant group x reverberation interaction for either the F0 or the F1 DL measures. Eyeballing the behavioral data in Figure 6 of their paper confirms that reverberation affects behavioral performance of musicians at least as much as that of non-musicians.
Lines 285-286, in reference to speech perception in babble noise.
Coffey et al. (2017) is a review paper that examined many other types of noise in addition to babble. It does seem to make a case for speech perception in noise. It also mentions that some studies did not find an effect. Interestingly, Coffey also mentions that "(Boebinger et al. 2015 for example calculated that subjects per group would have been required to achieve statistical power at the 0.8 level, which is much larger than most of the samples used in the reviewed studies)."
Parbury-Clark 2009 shows an effect in two tests but not in two other tests. The physical size of the effect in not reported numerically but the top panel of Fig. 1 looks like the difference is about 0.6 to 0.7 dB SNR. This is a small but meaningful difference. Slater et al. (2017) found a difference of 0.68 dB SNR favoring musicians. (p=0.033, N = 17 mus 14 non-mus). Summarizing these results as an advantage of about 1 dB SNR seems excessive.
***Strait et al. (2012) used two measures of SIN perception: WIN, and HINT (which was run in three sub-conditions, with noise on the left, front or right and the signal always coming from the front). There was no statistically significant difference between musicians and non musicians in either test. The 2-way ANOVA for HINT data, using condition and group as factors, reported no significant effect of group (musicians vs. nonmusicians)! Even when handpicking post-hoc comparisons, it was hard to find a significant result. The noise-in-front condition resulted in no significant difference. The musician/nonmusician comparisons for noise-left and noise-right are not reported, but when adding the noise-left and noise-right condition a significant difference is found. Also, it is worth mentioning that the outcome measure for the HINT measures in this study was percentile rankings rather than the more easily interpretable SNR, so it is hard to determine the practical significance of any difference. Furthermore, the only speech perception measure to show a musician/nonmusician difference (Figure 1A, left) shows that the average performance for musicians was about 50%, meaning they were close to the norming sample. The difference between groups arises because the nonmusicians perform significantly below the norms.
In summary, the Strait et al. study is perfectly consistent with the hypothesis that musicians do not understand speech in noise better than nonmusicians. The study is also a good demonstration of how some investigators in this area resort to cherry-picking results until a significant difference is found.
***Lines 289-290, “One intervention study has even shown that 2 years of musical training leads to improvement in speech in noise perception in normal hearing children, demonstrating a causal role of musical training (Slater et al., 2015).”
Again, results of this study are much less conclusive than what the original authors or the present review would have us believe. The design of the study is very nice, with a control group that received training only in year 2, allowing for the assessment of within- and between-group differences. The between-group comparison showed a statistically significant difference (p=0.022 for the group by year interaction), but the within-group comparison does not show a difference. I say this because for Group 1 there was no evidence of more improvement in Year 2 (music training) than in year 1 (no training). One bizarre experimental detail is that group 2 had 27 subjects but only 19 were randomly selected for comparison to Group 1. Last but not least, results from Holder et al. (Otology and Neurotology, 2016) show that normal hearing children from 5 to 12 years old improve their speech perception in noise by 0.52 dB per year for BKB-SIN and 0.79 per year for QuickSIN. It is reasonable to expect that SNR for HINT sentences would also improve with age, even in the absence of any intervention. It is strange that Group 1 in this study shows little to no change over two years (certainly no more than 0.2 dB or so over two years, with the second year under musical training). Given the strange results for Group 1, the advantage of group 2 over group 1 (and thus the between-group comparison) becomes more uncertain and, as we said at the beginning of the paragraph, the within group comparison did not show any effect of musical training.
***Lines 292-293, “It has been shown that children with CI in a musical environment have enhanced speech in noise (Torppa et al., 2018)”.
This is a misinterpretation of the Torppa et al., 2018 study. The difference between the two groups cannot be described as a difference in musical environment. There was no statistically significant difference in musical environment between the singing and the non-singing group. As the paper says, "Chi-square confirmed that the CI groups did not differ in the attendance at supervised musical or other activities outside of the home".
Also, there was no significant interaction of group with time. That is, there was no evidence the the singing group improved more than the non-singing group over time.
Finally, the singing group did better at speech perception in noise: "the CI singing group had better perception of speech in noise (lower SRT75) than the CI non-singing group (F1,19 = 4.98; B = 1.96, reference= CI singing group; p = .038) (Figure 4c)”. What this suggests is that, for some reason, children with cochlear implants who choose to sing in their everyday life tend to understand speech in noise better than children with cochlear implants who choose not to sing very often.
***Lines 293-294, “… as well as better auditory scene analysis after musical training (Rochette et al.)”.
This is another major misinterpretation of a study. The children who received musical training (N=14) attended their oral deaf school full time; those who did not receive musical training (N=14) attended half-time. None of them were evaluated before musical training, so it is misleading to say that one group was better "after musical training". For all we know, they were also better before the training. Also, there were differences other than musical training: full time vs. half time school attendance. Children were not randomized to each group. Lastly, these were not "children with CI". Less than 50% of the children used CIs.
***Lines 303-304, “In children with CI, one study showed that 6 months of piano lessons enhanced emotional prosody perception (Good et al., 2017).”
Good et al do claim that "Critically, music training also improved emotional speech prosody perception" but this is not true.
There were 9 children in each group. Despite the very small N, this study had two important strengths: a control group, and pre-training measures.
Tests of musical skill showed "an interaction between time and group [F(2,32) = 3.66, p = 0.037".
However, for prosody perception there was no main effect of group (musical vs art training) and no group by time interaction. In other words, musical training did NOT show statistically significantly greater improvement for the music training group than for the art training group.
The fact that "A main effect of time was found for the music group [F(2,15) = 5.6, p = 0.015] but not for the art group [F(2,15) = 1.58, p = 0.24]" is completely irrelevant, because any statistician worth his/her salt knows that the difference between a statistically significant result and a non-statistically significant result is not, by itself, statistically significant. In fact, I recommend reading Gelman and Stern (2006), "The Difference Between “Significant” and “Not Significant” is not Itself Statistically Significant", The American Statistician.
***Lines 314-316, “Musical training has been effective to restore syntactic processing in various clinical populations.” Kotz et al. (2005) is cited in support of this statement.
This is wrong. Kotz et al. simply did not measure syntactic processing. They only measured ERPs.
Furthermore, Kotz et al. could not have shown any behavioral effect of musical training (had they chosen to measure it) because no musical training was provided in their study. Rhythmic priming was provided to patients (N=9) and normal controls (N=9). There was no condition without priming, so the effect of priming could not be established. Even if we were to call three minutes of rhythmic priming "musical training", the Kotz study does not, and cannot, show the effect of that intervention, due to its experimental design.
***Lines 316-317, “Similarly, rhythmic primes have been beneficial for grammaticality judgment in children with specific language impairment (Bedoin et al., 2016; Przybylski et al., 2013).”
This statement is misleading because rhythmic primes seem to be beneficial for grammaticality judgments conducted shortly after priming, across the board, regardless of whether a child has SLI or not. Bedoin et al. report that "the main effect of musical prime was significant, F(1, 30) = 4.92, p = 0.03” and they also report that it did not interact with group, p = 0.32. In other words, it's misleading to say that rhythmic primes were beneficial for grammaticality judgment for SLI kids: they were beneficial for everyone and there was no priming-group interaction.
Same exact result for Przybylski et al: there was a significant effect of priming and there was no priming-group interaction.
A separate issue is that, from a practical standpoint, many of these studies use priming and the grammaticality stimulus presentation takes place a few minutes later. Is there a leftover effect a day later? If not, priming would not be very useful as therapy.
***Line 319, “… a positive, although moderate, effect of rhythmic primes on both syntax and grammar (Bedoin et al., 2018).”
The crossover design was a nice aspect of this small (N=10) study.
In any case, the statement is only partially correct. The primes did have an effect on grammaticality judgements (“Priming x Testing interaction [F(1,9) = 4.86, P = 0.05”) but no significant interaction term was reported for syntactic comprehension, nonword repetition, two measures of visual selective attention, or memory. More specifically, primes did not have any more effect than the neutral, non-musical environmental sounds for the purpose of syntactic comprehension.
Last but not least, it is not clear to me, that the papers chosen for review were selected in an unbiased manner. For example, and with respect to the specific issue of speech perception in noise by musicians and nonmusicians, a quick search came up with several studies that did not find an effect and were not cited in the present review:
https://www.ncbi.nlm.nih.gov/pubmed/24489819
https://www.ncbi.nlm.nih.gov/pubmed/28974705
https://www.ncbi.nlm.nih.gov/pubmed/31320656
https://www.ncbi.nlm.nih.gov/pubmed/25618067
as well as one that was just published in Ear & Hearing:
https://journals.lww.com/ear-hearing/Abstract/2020/03000/The_Effect_of_Musical_Training_and_Working_Memory.7.aspx
(although the fact that this study controlled for working memory leaves open the possibility that an effect of musical training would be present, if the authors chose NOT to control for working memory).
Reviewer 2 Report
Title: Musical Training for Auditory Rehabilitation in Hearing Loss
The authors address an important problem associated with rehabilitation of severely-to-profoundly deaf with cochlear implants (CIs). They present an opinion paper reviewing the benefits of musical training in the rehabilitation of hearing loss. This assessment is not new and has been proposed previously. Overall, the paper is a well written review on the correlation between musical training and performance with rehabilitative measures for hearing loss. The key literature has been referenced. The enthusiasm of this reviewer for the paper has been dampened by the fact that the focus is only on the benefits of musical training as a key factor enhancing language perception and production. The authors state that early rehabilitation techniques focus on low-level sensory abilities without taking into consideration that cognitive operations are important for auditory perception. The authors then point out that musical training is an appealing candidate for such an approach. While they are correct with their statement they omit a great opportunity to contrast music training against other approaches for rehabilitation with similar results. A critical evaluation of the papers cited is missing. Furthermore, one would have expected that other factors, other than music training that correlate with similar enhancements would have been discussed.
Round 2
Reviewer 1 Report
I appreciate the authors’ constructive response. I think the manuscript has improved, but I still see a few spots that could be improved without much difficulty.
I still disagree with the wording “Overall, there has been a massive accumulation of evidence in the last decades pointing to an improvement induced by music training at different levels of speech and language processing (Schon & Morillon, 2018), although a debate on these effects remains open especially at the high processing levels.” As we have seen, a large number of studies in this area are poorly designed, the data are incorrectly analyzed, or the conclusions are highly speculative and do not follow from the data. I propose the following wording:
“Overall, there has been an increasing number of studies in the last decades pointing to an improvement induced by music training at different levels of speech and language processing (Schon & Morillon, 2018), although a debate on these effects remains open especially at the high processing levels.”
I still believe that the Przybylski et al. study has been completely mischaracterized. You say that the study “has demonstrated the positive effect of rhythmic priming on dyslexic children” and this is patently not true. I repeat, the statement is just not true. As I said in my initial review “the only observed effect was that regular musical prime sequences had a positive effect on grammaticality judgments with respect to irregular musical prime sequences. We do not know what performance would have been obtained in the absence of musical priming.” That’s it. After reading the Przybylski et al. paper all we can conclude is that one intervention was better than another intervention, but we just don’t know whether any of the interventions are better or worse than no intervention.
You defend your statement by saying “Later in the manuscript we cite other works by the same group and more precisely Bedoin et al. (2016) who showed that the effect is indeed a benefit” and this precisely proves my point. Other studies may show that, but this study in particular does not, so please do not say that it does.
Please address this mistake, which should be trivially easy to do.
Finally, I have a couple of comments that don’t require any action on your part. You propose an interesting hypothetical example related to two treatments for a disease. You ask a question: “However, treatment A has shown significant recovery, while the other (B) did not. What would you suggest to your friend: A or B?”. The answer is not as obvious as you might think because the word “significant” in your question actually means “statistically significant”. Let’s say that the illness in question is cancer and that we know for a fact, with absolute certainty, that my friend’s life expectancy is exactly 1 year. Let’s assume that the 95% confidence interval for how long my friend will live with treatment A is [1.01-1.03] years. This means that treatment is statistically significantly superior to no treatment (p<0.05). Let’s now assume that the 95% confidence interval for how long my friend will live with treatment B is [0.99-15] years. This means that treatment B is not statistically significantly superior to no treatment (p>0.05). Under these circumstances I would strongly recommend treatment B to my friend.
Lastly, I appreciate the gentle joke about the papers I selected at the end of the review to make my point. Those papers were not, indeed, intended to be a representative sample of published studies about speech perception in noise by musicians. They are not. I must say I feel I don’t know whether musicians understand speech in noise better than nonmusicians. I wish somebody did a large-scale, preregistered study, to answer the question once and for all.
In any case, thanks again for your thoughtful response to my review. I look forward to seeing your paper in print.
Reviewer 2 Report
The authors have addressed my suggestions and comments.
Author Response
We thank the reviewer for his/her work to improve our manuscript.